# Gene Variant Related Neurological and Molecular Biomarkers Predict Psychosis Progression, with Potential for Monitoring and Prevention

**DOI:** 10.3390/ijms252413348

**Published:** 2024-12-12

**Authors:** Stephanie Fryar-Williams, Graeme Tucker, Peter Clements, Jörg Strobel

**Affiliations:** 1Youth in Mind Research Institute, Unley, SA 5061, Australia; 2Department of Medical Specialities, Adelaide Medical School, Faculty of Health and Medical Sciences, The University of Adelaide, Adelaide, SA 5000, Australia; 3Department of Public Health, Adelaide Medical School, Faculty of Health and Medical Sciences, The University of Adelaide, Adelaide, SA 5000, Australia; 4Department of Paediatrics, Adelaide Medical School, Faculty of Health and Medical Sciences, The University of Adelaide, Adelaide, SA 5000, Australia; 5Department of Psychiatry, Adelaide Medical School, Faculty of Health and Medical Sciences, The University of Adelaide, Adelaide, SA 5000, Australia

**Keywords:** *MTHFR* C677T, methylation, schizophrenia, prognosis, disease progression, risk prediction, sensory processing, suicide, hostility, bipolar, treatment resistance

## Abstract

The (*MTHFR)* C677T gene polymorphism is associated with neurological disorders and schizophrenia. Patients diagnosed with schizophrenia and schizoaffective disorder and controls (*n* 134) had data collected for risk factors, molecular and neuro-sensory variables, symptoms, and functional outcomes. Promising gene variant-related predictive biomarkers were identified for diagnosis by Receiver Operating Characteristics and for illness duration by linear regression. These were then analyzed using Spearman’s correlation in relation to the duration of illness. Significant correlations were ranked by strength and plotted on graphs for each *MTHFR* C677T variant. Homozygous *MTHFR* 677 TT carriers displayed a mid-illness switch to depression, with suicidality and a late-phase shift from lower to higher methylation, with activated psychosis symptoms. *MTHFR* 677 CC variant carriers displayed significant premorbid correlates for family history, developmental disorder, learning disorder, and head injury. These findings align with those of low methylation, oxidative stress, multiple neuro-sensory processing deficits, and disability outcomes. Heterozygous *MTHFR* 677 CT carriers displayed multiple shifts in mood and methylation with multiple adverse outcomes. The graphically presented ranked biomarker correlates for illness duration allow a perspective of psychosis development across gene variants, with the potential for phase of illness monitoring and new therapeutic insights to prevent or delay psychosis and its adverse outcomes.

## 1. Introduction

Schizophrenia and schizoaffective disorder are progressive disorders with considerable comorbidity and overlap [1]. They have a 1 to 4% incidence in the general population [2] and may arise during adolescence or adulthood associated, with the complex interaction of genetic, environmental, hormonal, cognitive, and oxidative stress and a range of neurological, developmental, or autistic features [3,4,5,6,7,8,9]. Due to the brain’s struggle to functionally integrate information in a setting of multiple contributing factors, presenting symptoms of psychosis may incubate in a subthreshold manner for some time, before overt symptoms reveal a clear diagnosis [10]. Although symptoms resolve in some instances, two-thirds of cases progress, with a growing number of symptoms of increasing intensity and disability [11]. No objective biomarker progression pathway has yet been established for psychosis development, and diagnostic methods remain reliant on the systematic interpretation of symptom-behavior complexes [12].

The developmental nature of schizophrenia is supported by neuroimaging studies that detect neurological changes from first-episode psychosis to chronic schizophrenia [13,14]. Furthermore, mounting evidence indicates that schizophrenia is not a singular condition but part of a spectrum with bipolar affective disorder [15,16,17]. Moreover, there is evidence of considerable variability in clinical diagnostic allocation over time [18,19,20]. Adding to this uncertainty are contradictory findings about the issue of treatment for early psychosis symptoms, as antipsychotic treatment appears to prevent later brain changes [21], but may exacerbate brain shrinkage over time [22]. These neurological considerations create a therapeutic dilemma since it appears that delayed treatment may contribute to a poorer outcome [23]. For these reasons, it is crucial to dissect and identify the components of psychosis progression, allowing the neurobiological underpinnings of psychosis-linked conditions to be monitored as symptoms evolve over time [24]. This approach enables the development of therapeutic interventions that are precisely tailored to specific illness phases and biomarker changes.

The homozygous (TT) polymorphism at position 677 of the methylene tetrahydrofolate reductase (*MTHFR*) gene has a well-known association with developmental neurological defects, such as spina bifida [25], and the role of this gene polymorphism in schizophrenia has been established by meta-analysis [26,27].

Our investigation of *MTHFR* C677T gene variants in relation to biomarkers for schizophrenia and schizoaffective disorder yielded a range of highly predictive biomarkers obtained by linear regression modeling of the data for both raw variables and variables that had been determined as diagnostically significant on Receiver Operating Curve (ROC) curve statistics [28]. Several clusters of neuro-sensory processing disorder, molecular abnormalities, and risk factors for schizophrenia and schizoaffective disorder, emerged from this statistical analysis. Several biomarkers from these clusters were highly predictive of both diagnosis and duration of illness (DOI) [28]. From our past translational research analysis, we were already aware of the relationship between molecular biomarkers and neuro-sensory processing disorders [29]. We were also able to construct a functional molecular landscape from which these biomarkers were selected, which explained the relationship between molecular mechanisms and *MTHFR* C677T gene variants. This work also reviewed research evidence and common biochemical knowledge regarding molecular interactions within methylation pathways [30]. Importantly, molecular and neuro neuro-sensory clusters that were >92% predictive of duration of illness associated with the *MTHFR* 677 CC variant, represent a low methylation state and these biomarkers are also predictive of adverse functional and social outcomes [28].

In the clinical setting, it is useful for clinicians to have some appreciation of factors influencing an individual’s duration of illness (DOI), as such knowledge can reduce symptoms and greatly assist therapeutic planning. We believe that analyzing our diagnostic biomarkers in relation to illness duration (DOI) might provide more information in this regard. Our aim was to provide insights into which biomarkers could best predict the duration of illness and to visualize these biomarkers in terms of the course of illness. We particularly wished to understand which biomarkers might precede a diagnosis as well as influence the course of illness progression, and to discover which of these biomarkers could be remediated early in the illness trajectory to offset or delay psychosis progression.

### 1.1. Relationship of the MTHFR Enzyme to the MTHFR C677T Gene

The homozygous (TT) variant is a single nucleotide polymorphism (SNP) mutation with different SNP phenotype codes (OMIM:607093, NM_005957.4(*MTHFR*): c.665C>T, p.Ala222Val, rs1801133) in different genotype classification systems. It has an autosomal recessive inheritance pattern and an estimated 8.5% prevalence in the Australian population [31,32,33,34], but varies according to ethnicity and geographic locality [32,34,35]. It has been linked to adverse birth outcomes, birth defects, pregnancy complications, adult cardiovascular disease, and schizophrenia [36,37,38]. Its role in methyl group donation makes it a key epigenetic player in DNA synthesis and DNA methylation [39], and research data suggest that this is a gene-specific methylation effect rather than a demethylation effect [40]. In contrast, the homozygous wild-type *MTHFR* 677 CC variant codes for a normally active MTHFR enzyme that is theoretically able to produce sufficient 5-MTHF for the continuation of the methylation cycle by the metabolism of homocysteine to methionine [41]. The third heterozygous *MTHFR* 677 CT variant was also investigated in this study. Figure 1 demonstrates the pathways closely related to the MTHFR enzyme, and further description of these pathways is available in previous research [28,30]. This MTHFR enzyme is encoded by three different *MTHFR* C677T gene variants. These variants are denoted as TT, CT, and CC according to the presence of either Thymidine or Cytosine at the 677 sites of the C677T gene on exon 4. When the nucleoside Cytosine is replaced by Thymidine (T) at this site, there is a transition from alanine to valine [Ala22Val] in the catalytic region of the encoded enzyme. This transition results in a thermolabile, dysfunctional enzyme that dissociates away from its riboflavin—related FAD cofactor and is unable to produce 5-methyltetrahydrofolate (5-MTHF). This effect is associated with increased FAD cofactor availability for other enzymes and a 50–75% reduction in MTHFR enzyme activity compared with the activity of the unaffected *MTHFR* 677 CC variant [31,33].

### 1.2. Reasoning and Selection of Candidate Markers in This Study

A Legend of Abbreviations for biomarkers and other substances related to this study is provided at the beginning of the Appendix A. Specific biomarkers assayed in this study and the methods employed are presented in Appendix A, along with method citations. They were selected according to their accessibility for commercial testing as well as their role as cofactors in the biochemistry pathways of methylation, as outlined in (Figure 1). The MTHFR enzyme is largely influenced by vitamin B2 (riboflavin), which is a precursor of its flavin dinucleotide (FAD) cofactor. Other enzyme cofactors include the activated forms of folate, vitamin 12, vitamin B6, and vitamin D. Levels of trace elements copper and zinc, histamine, and a marker for oxidative stress have also been assayed [42]. Figure 1 also demonstrates remote pathways regulated in synchrony with methylation, in which the amount of riboflavin-activated vitamin B6 cofactor availability varies with riboflavin availability. The riboflavin-activated vitamin B6 cofactor and flavin cofactors such as methyl folate contribute to the synthesis or metabolism of important intermediate molecules such as serine, tryptophan, glycine, and histamine [43,44,45,46] in multiple peripheral pathways [47,48].

### 1.3. Relationship of MTHFR Enzyme to Methylation Cofactor 5-MTHF, Catecholamine Levels

As briefly described in Section 1.1, the MTHFR enzyme produces a methylated, functionally activated form of folate, named 5-methyltetrahydrofolate (5-MTHF), which 5-donates one carbon to homocysteine to form methionine. This allows methionine to generate the ubiquitous methyl donor, S-adenosyl methionine (SAMe) [49]. SAMe contributes to RNA and DNA methylation and, therefore, contributes to the epigenetic regulation of gene expression [50]. SAMe plays a cofactor role in histamine metabolism [51,52] and is a necessary cofactor for the catechol-o-methyl transferase (COMT) metabolism of catecholamines [53,54]. Furthermore, SAMe serves as a cofactor, allowing the conversion of noradrenaline [NA] to adrenaline [AD] [55]. 5-MTHF also activates vital vitamin cofactors, such as vitamin B12 [56]; therefore, the absence of such activation has multiple adverse effects in many allied enzyme reactions. To counteract the loss of 5-MTHF and the inability to convert homocysteine directly to methionine, carriers of the *MTHFR* 677 TT variant can metabolize homocysteine to methionine using an alternative pathway that operates across the methylation cycle. In this alternative pathway, the enzyme betaine hydroxy methyltransferase (BHMT) uses zinc as a cofactor and trimethyl glycine (TMG) as an alternative methyl donor to 5-MTHF [57,58,59]. Over-compensatory activation of this BHMT enzyme may result in excessive methylation and high levels of SAMe generation in a process called over-methylation. In this process, BHMT utilizes zinc, so zinc may become depleted in the plasma and serum compartments. In the serum, zinc exists in a reciprocal relationship with copper, which is regulated by its binding to serum ceruloplasmin [60]. Accordingly, free zinc and copper, expressed as a ratio of free zinc to copper, emerges as a biomarker in this study. Indeed, zinc and copper play key roles as enzyme cofactors in multiple allied pathways related to dopamine metabolism and neuronal function [61,62].

### 1.4. Relationship of the MTHF Enzyme to Flavin Cofactors and Methylation

Flavin adenine nucleotide (FAD) and its precursor flavin mononucleotide (FMN), are derived from riboflavin (vitamin B2) in food and synthesized by various gastrointestinal organism strains of *Bacillus subtilis*, *Lactobacilli species*, *Escherichia coli*, and *Saccharomyces cerevisiae* [63,64,65]. Through its attachment to the MTHFR enzyme, FAD is a necessary cofactor for 5-MTHF production and, thereby, for the activation of several vitamins related to methylation, folate, and catecholamine pathways. These vitamins are vitamin B6, vitamin B12, and vitamin D [41,43,44,45,46,47,48,66,67,68]. For these reasons, the selection of the activated form of biomarkers was a priority for this project, and we consider that many of the low vitamin cofactor outcomes associated with the *MTHFR* 677 CC and CT variants could be explained by dysfunctional mitochondrial function or low microbiome populations related to methylation and low riboflavin precursor for synthesis of FMN and FAD [64,69,70].

### 1.5. Relationship of Flavins to Catecholamines

As a cofactor for the MTHFR enzyme, FAD assists in the synthesis of the active form of folate, 5-methylene tetrahydrofolate [5-MTHF]. Therefore, in a setting of low vitamin B2 availability and low FAD, there is low reconstitution of methionine and, consequently, low synthesis of S-adenosyl methionine [SAMe] from its precursor, methionine [71]. Since SAMe is a necessary cofactor for catechol-o-methyltransferase (COMT) metabolism of catecholamines [53,54], and since FAD acts to cofactor the monoamine oxidase [MAO] enzyme [72], interruption of SAMe synthesis together with low FAD availability, compromises catecholamine metabolism at three levels of MAO, COMT and conversion of noradrenaline [NA] to adrenaline [AD] [55], leading to conserved, unmetabolized dopamine (DA) and noradrenaline (NA). Also, SAMe is required for histamine metabolism by the enzyme histamine methyl transferase [HMT] [51,52] and for creatine formation from glycine [73,74], therefore each of high histamine levels and low creatinine levels characterize a low methylation state with reduced SAMe synthesis.

### 1.6. Relationship of the MTHFR Enzyme to Vitamin B6 and Oxidative Stress

FMN-activated vitamin B6 has been shown to have a synergistic relationship with visual and auditory neuro-sensory processing disorders and elevated catecholamines in schizophrenia [29], and this relationship has been shown to vary with *MTHFR* 677 variant [28,75]. It has been previously reported that vitamin B6 is activated in a manner that is affected by the availability of MTHFR enzyme precursor flavin molecules [47,48]. In the methylation cycle, activated vitamin B6 is required to cofactor the S-adenosylhomocysteine hydrolase enzyme (SAHH), which is the enzyme downstream of SAMe that produces S-adenosyl homocysteine (SAH), from which molecule, homocysteine is eventually produced. Although homocysteine is forward metabolized to methionine, by methionine synthase (MS), it is also metabolized by the trans-sulphuration (TSF) pathway [76] (Figure 1). Here, homocysteine joins with serine to contribute to the synthesis of the body’s major antioxidant glutathione (GSH) [77]. This synthesis is undertaken by the vitamin B6-dependent enzyme cystathionine beta transferase (CBS), which is inhibited by copper [78,79]. This means that when copper is high because zinc is low (as happens when the BHMT enzyme is compensating for *MTHFR* 677 TT enzyme’s low methylation ability), the homocysteine precursor for this process is conserved at the expense of glutathione production. Glutathione (GSH) is a major redox-regulating agent in cells, and in the absence of its active, reduced form, unquenched free radicals damage mitochondrial function, producing an intracellular state of oxidative stress. Oxidative stress is an acknowledged component of schizophrenia [80]. Also, after glutathione has performed its reductive function, FAD is a necessary cofactor for the reconversion of free radical oxidized glutathione [GSSH] back to its active, reduced form of GSH [81].

### 1.7. Relationship of MTHFR 677 Variants to the Neurology of Neuro-Sensory Processing

Six neuro-sensory processing variables (Appendix A) were examined in this study to ascertain how these disturbances might relate to the progression of psychosis over the duration of illness. Impairment in working memory is a well-documented finding in schizophrenia [82,83], while clinical experience combined with research suggests that both auditory and visual end-organ dysfunction contribute to the cognitive impairment detected in schizophrenia [84,85]. Further research has indicated that deep intra-cerebral white matter abnormalities in the corpus callosum disconnect intra-cerebral processing [86]. For these reasons, the selection of neuro-sensory markers in this study was devised to span both auditory and visual domains and cover both frontal-parietal and deeper cerebral circuitry [87]. Within this circuitry, working-memory utilizes transverse cortical neuronal tracts and dichotic listening (CW age diff %) parameters found to be abnormal in this study, indicate that auditory information processing is disturbed in white matter tracts of the corpus callosum [88].

## 2. Results

### 2.1. Duration of Illness Analysis for MTHF C677T Variants

The maximum DOI for the *MTHFR* 677 CC carriers was 44 years *(n* 60, mean 7.18 years SE 10.5). The maximum DOI for the *MTHFR* 677 TT variant carriers was 24 years *(n* 7, mean 5.7 years, SD 9.4). The *MTHFR* 677 CT variant carriers had the longest DOI of 47 years (*n* 61, mean 9.05, SD 10.5) (Appendix A).

### 2.2. Receiver Operating Characteristics Results for MTHFR C677T Variants

Receiver Operating Characteristics (ROC analysis) results for each of the *MTHFR* C677T variants with respect to the diagnosis of schizophrenia or schizoaffective disorder can be found in a previous publication [28] and in Appendix A. On ROC analysis, a variable is given the status of a “biomarker” according to the distinctive parameter of Area Under the Curve (AUC). This is accompanied by parameters of Odds Ratio (OR), Sensitivity, Specificity, and Positive and Negative Predictive Value. Further analytical information about the AUC parameter is provided in the Methods Section of this manuscript (Section 4.5), and biomarker abbreviations are repeatedly given in this publication, as well as presented on Appendix A. Also, the significance of many selected molecular marker variables has already been outlined in the Introduction Section 1 and is available in previous Literature Reviews and Research articles [28,29].

Diagnostic biomarkers discovered by ROC analysis for the *MTHFR* 677 CC variant (*n* 65, prevalence 48.5%) are provided in Appendix A. These biomarkers are mainly characteristic of a lower methylation state, with many neuro-sensory processing deficits and adverse functional outcomes. ROC analysis for this variant also revealed significant premorbid risk factors for diagnosis, such as a family history of mental illness and a history of abuse. Other risk factors of significance in ROC analysis are a history of ear infections, developmental disorder, learning disorder, and head injury. These factors are mainly aligned with low methylation markers as well as biomarkers for oxidative stress, suicidality, hostility, multiple neuro-sensory processing deficits, increased illness severity, and low functional outcomes. In this context, notable ROC-identified molecular biomarkers were found for high levels of 5 hydroxy indole acetic acid (5-HIAA), dopamine, and 5-HIAA combined (DA × 5-HIAA), elevated noradrenaline (NA) to dopamine ratio (NA/DA), and elevated NA/MHMA ratio (where MHMA is 3-methoxy-4-hydroxymandelic acid is a NA and adrenaline (AD) metabolite), (Figure 1). Other ROC-identified diagnostic biomarkers were for elevated AD/MHMA ratio, elevated histamine, and the oxidative stress marker hydroxyhemopyrroline-2-one (HPL), [42]. In contrast, low levels of biomarkers were found for red cell folate, vitamin B6, and vitamin D. Hostility and suicidality symptoms were also identified as significant biomarkers of this *MTHFR* 677 CC variant (Appendix A), and ROC analysis identified six neuro-sensory biomarkers associated with the diagnosis. These biomarkers represented delay in both auditory and visual processing, delay in auditory and visual working memory, and dichotic listening deficit.

In contrast, diagnostic biomarkers discovered on ROC analysis for the homozygous *MTHFR* 677 TT variant (*n* 7, prevalence 5.2%) (Appendix A) were far less numerous, and most of the low methylation state molecular biomarkers characteristic of the *MTHFR* 677 CC variant were absent. In contrast, there were molecular variables indicating the presence of a high methylation state related to the schizophrenia diagnosis, for carriers of the *MTHFR* 677 TT variant. Notably, AD levels exceeded NA levels in the AD/NA ratio, implying the availability of a higher level of methylated cofactor (SAMe) for the conversion of NA to AD. A further finding was the high ROC significance for vitamin B2/creatinine, dopamine (DA)-free copper to zinc ratio (Cu/Zn), and the oxidative stress marker hydroxyhemopyrroline-2-one (HPL) [42]. However, neuro-sensory biomarkers emerging toward the end of the illness trajectory, demonstrated that in the final illness phase, *MTHFR* C677 TT carriers did not fare much better than *MTHFR* 677 CC carriers. There may be different reason for this, because findings of a very strong ROC for abnormal otoscopy in association with the TT variant, indicate the presence of middle ear disease. In this setting, primary ear conduction pathology could be expected to influence other neuro-sensory parameters, such as dichotic listening (competing words % difference ROC) and reduced auditory processing speed (reduced auditory speed of processing ASOP % ROC). In addition, it appears that visual neuro-sensory processing systems are also impaired, with impaired visual acuity (distance vision ROC) contributing to reduced visual speed of processing (VSOP age add % ROC). In this setting of a mixture of visual and auditory end organ dysfunction and significant intra-cerebral sensory processing delay, collateral symptoms of confused thinking and hostility are not unexpected findings. Indeed, these symptoms yield extremely significant ROC’s, contributing to the high symptom intensity index (SIR), in carriers of the *MTHFR* 677 TT variant, (Appendix A).

Since the *MTHFR* 677 CT variant (*n* 62, prevalence 46.3%) has one C and one T allele, it was not unsurprising to find ROCs for biomarkers representing both low and high methylation states (Appendix A). The ROC biomarkers showed a metabolic bias toward the lower methylating profile (that is characteristic of the *MTHFR* 677 CC variant). For instance, there were strong lower methylation signatures of higher NA/MHMA ROC and accompanying high ROCs (and Odds Ratios) for developmental and learning disorders. However, in keeping with expectations of the T allele having some influence on the outcome, there were hints of a concurrent shift to a high methylation state, represented by an ROC for the characteristic signature of adrenaline levels exceeding noradrenaline levels (AD/NA ROC, *n* 61, AUC 0.6522, *p* 0.0030). These ROCS, representing contrasting methylation states, were accompanied by a significant number of ROCs for compound biomarkers, representing a mixed methylation process [29]. Strong ROCs for developmental disorder (*n* 62, AUC 0.668, *p* 0.000) and learning disorder (*n* 62, AUC 0.677, *p* 0.006) accompanied these findings. In this disordered developmental setting, with ROC identified biomarkers representing bivalent methylation states, it is not unexpected that *MTHFR* 677 CT carriers experienced the full number of assessed sensory processing deficits (Appendix A). These are expected to contribute to intra cerebral sensory processing confusion that accompanies learning disorder which leads on to psychosis and the significant ROC results for suicidality (*n* 61, AUC 0.6129, *p* 0.0016) and hostility (*n* 61, AUC 0.871, *p* 0.000), with high symptom intensity (SIR) and global disability (GAF).

### 2.3. MTHFR 677 CC Variant Linear Regression Results for Duration of Illness

On linear regression analysis for DOI, the *MTHFR* 677 CC variant (*n* 60, prevalence 48.5%, maximum DOI 44 years, mean DOI 7.18 years) yielded a predictive model containing molecular biomarkers that were characteristic of a low methylation state (NA, DA, AD/MHMA, DA × 5-HIAA). These variables are presented in Appendix A and summarized in Table 1, with their mechanistic meaning explained in previous articles [28,29] and in the Introduction Section 1 and Discussion Section 3 of this manuscript.

### 2.4. MTHFR 677 TT Variant Regression Results for Duration of Illness

On linear regression analysis, three DOI-related predictive biomedical markers for *MTHFR* 677 TT variant polymorphism (*n* 7, prevalence 5.2%, maximum DOI 24 years, mean DOI 5.7 years) are found in Table 2 and Appendix A. They rule out any vitamin B2 or B6 deficits through the finding of that DOI is positively predicted by high levels of vitamin B2 and vitamin B6, alongside a predictor with negative valence for dopamine (DA) combined with the serotonin metabolite 5-HIAA. These findings may link the role of vitamin B2 and vitamin B6 in serotonin metabolism, as explained in previous articles [28,29,89] in the Discussion Section 3.4 of this manuscript.

### 2.5. MTHFR 677 CT Variant Linear Regression Results for Duration of Illness

As expected from the presence of mixed C and T alleles in the heterozygous *MTHFR* 677 CT variant (*n* 60, prevalence 46.2%, maximum DOI 47 years, mean DOI 9.05 years), linear regression results show a mixture of low and high methylating signatures. These are summarized in Table 3 and presented in detail in Appendix A. The mixed methylation predictor vitamin B12/activated vitamin D is a good compound biomarker signature for carriers of this heterozygous *MTHFR* 677 CT variant, and its meaning has been explained in previous articles [28,29] and in Discussion Section 3.5 of this manuscript.

### 2.6. MTHFR 677 CC Graphing of Ranked Correlates for Duration of Illness

When Spearman’s correlates were determined for predictive ROC and linear regression biomarkers for the *MTHFR* 677 CC variant, and correlates were ranked from least to strongest (Appendix A), correlates could be plotted on a virtual timeline graph, according to the method described in Section 4.6. In this manner, psychosis progression was visualized for the *MTHFR* 677 CC variant (Figure 2), and several illness phases were observed according to the characteristic methylation states. Biomarkers aligned with these methylation states and different mental states were recognizable from discrete symptom clusters.

Phase 1 (white) displayed mounting abnormal biochemical characteristics of low methylation, with low vitamin cofactors, high 5-HIAA (indicative of serotonin metabolism), and high catecholamine levels (indicative of low methylation). These biomarkers occurred in a setting of attention impairment and auditory and visual processing deficits, leading to early dissociation symptoms, representing broken attentional flow and thought stream decoherence. These features were seen to be preceded by risk factors of a family history of mental illness, subclinical head injury, and a history of ear infections, together with lowered auditory memory, oxidative stress, and metabolic dysfunction.

Phase 2 (green) was characterized by frank symptoms of dissociation, such as experiencing blank episodes, accompanied by obsessional symptoms, such as mannerisms. There were also vitamin deficits indicated by lower correlates of vitamin B2 ROC, lower folate, lower vitamin D. Higher histamine levels were also present because a methylation (SAMe) deficit reduces histamine metabolism (Figure 1). Within this phase, Diabetes type 2 was also significantly related to DOI (*n* 58, *rho* 0.316, *p* 0.016).

Phase 3 (pink) had an onset of higher dopamine levels and was briefly associated with manic symptoms, such as elated mood, motor hyperactivity, and grandiosity, accompanied by lower vitamin B6 inhibiting tryptophan metabolism, with expected tryptophan trapping and increased serotonin degradation, leading to an overflow of 5-HIAA in urine.

Phase 4 (blue) had characteristics of cortical inhibition and was associated with depressed mood, self-neglect, motor retardation, learning disorder, and suicidality. The accompanying biomarkers were characterized by a low methylation state, represented by unmetabolized high catecholamine levels.

Phase 5 (pink) demonstrated an attempt to rebound from cortical inhibition (in a similar manner to post-dissociation rebound phase 2). This rebound was accompanied by a strong biosignature for noradrenaline (NA), which has a known association with high levels of arousal [90]. This phase was accompanied by arousal symptoms of anxiety, hostility, uncooperativeness, hallucinations, and bizarre behavior. In this phase of the graphic landscape, there was a disturbing convergence between the symptoms of hostility and impaired abstract thinking, with delayed auditory and visual working memory occurring alongside the biomarker for developmental disorder.

Phase 6 (white). This end phase was characterized by diagnostic certainty being finally reached. Unfortunately, this was not attained until frank psychotic symptoms emerged late in the virtual illness trajectory. By this time, early-warning risk factors and low methylation biomarkers had been in evidence across a maximum of 44 years (mean duration 7.18 years) of the virtual timeline. This allows ample time for preventative interventions to offset the final progression to full psychosis—a transition that is characterized by suspicious thought, impaired judgment, reduced insight, and maximal social and occupational disability (SOFAS).

### 2.7. MTHFR 677 TT Graphing of Ranked Correlates for Duration of Illness

In Figure 3, the homozygous *MTHFR* 677 TT variant displays a virtual timeline of illness development, where biomarkers differ markedly from that of the CC variant. Data relating to these correlates may be found in (Appendix A). Early on the virtual timeline, there is a relative absence of biomarkers; however, midway along on the timeline, DOI-related correlates suddenly appear for suicidality (*n* 7, *rho* 0.676, *p* 0.096). This concerning symptom is accompanied by symptoms of depressed mood (*n* 7, *rho* 0.737, *p* 0.059) and, uncooperativeness (*n* 7, *rho* 0.764, *p* 0.046) alongside biomarkers for dopamine (DA), (*n* 7, *rho* 0.791, *p* 0.034) and noradrenaline (NA), (*n* 7, *rho* 0.791, *p* 0.034) and the oxidative stress biomarker (HPL/creatinine), (*n* 7, *rho* 0.808, *p* 0.028). This combination of biomarkers is characteristic of conserved, unmetabolized DA and NA and reduced glutathione production in a low methylation state, as described in Section 1.5 and Section 1.6. These biomarkers are followed by a cluster of negative symptoms. These consist of social avoidance (*n* 7, *rho* 0.836, *p* 0.019), passivity and apathy (*n* 7, *rho* 0.836, *p* 0.019), lack of spontaneous conversation (*n* 7, *rho* 0.836, *p* 0.019), emotional withdrawal (*n* 7, *rho* 0.836, *p* 0.019) and self-neglect (*n* 7, *rho* 0.872, *p* 0.010). In this setting, there is also early reduction of Global Assessment of Function (GAF), (*n* 7, *rho* −0.777, *p* 0.040), clinical impression of high illness severity (CGI), (*n* 7, *rho* 0.794, *p* 0.033) and reduced social and occupational functioning (SOFAS), (*n* 7, *rho* −0.818, *p* 0.025).

In contrast to this sudden onset of mid-trajectory, mainly negative symptoms, later emerging symptoms are of cognitive disorganization (*n* 7, *rho* 0.956, *p* 0.001), suspiciousness (*n* 7, *rho* 0.957, *p* 0.001), anxiety *(n* 7, *rho* 0.923, *p* 0.003), hallucinations (*n* 7, *rho* 0.989, *p* 0.000) and hostility (*n* 7, AUC 1.000, *p* 0.000), with further discussion of hostility in Section 3.7. These symptoms emerge, alongside biomarkers implying a switch to high methylation, one feature of which is high vitamin B2/creatinine ROC (*n* 6, *rho* 0.908, *p* 0.000), as described in Section 3.4. Although DA ROC correlates of *n 7*, *rho* 0.956, *p* 0.001 arise in this setting, there are also prominent biomarkers representing high catecholamine turnover (HVA/DA), (*n* 7, *rho* 1.000, *p* 0.000) and (MHMA/NA + AD), (*n* 7, *rho* 1.000, *p* 0.000). Also, high correlation for abnormal otoscopy (*n* 7, *rho* 0.956, *p* 0.001) arises, indicating inner ear pathology, with delay in auditory speed of processing (*n* 7, *rho* 1.000, *p* 0.000) and there is also a biomarker for reduced visual span (*n* 7, *rho* 1.000, *p* 0.000).

Taken together, the DOI trajectory for the *MTHFR* 677 TT variant shows biomarker of sufficient early methylation without any abnormal symptoms, with mid-phase transition to negative symptoms with depression and biomarkers of low methylation. Finally, there is late phase transition into disorganized, activated psychotic symptoms with hallucinations and hostility. As discussed in Section 3.4, this DOI trajectory may represent a manic-depressive illness that develops over the duration of illness of carriers with the *MTHFR* 677 TT variant.

### 2.8. MTHFR 677 CT Graphing of Ranked Correlates for Duration of Illness

In keeping with the overlap or competition effects from the two C and T allele components, the heterozygous *MTHFR* 677 CT variant demonstrated a notably unstable trajectory, with an early phase of high methylation, followed by a swing to a low methylation state. Data relating to this graph are found in Appendix A, and the trajectory is shown in Figure 4. There is a marginal risk of early suicidality (*n* 61, *rho* 0.215, *p* 0.097) accompanied by a history of family mental illness, history of abuse (*n* 61, *rho* 0.287, *p* 0.025), and head injury (*n* 62, *rho* 0.341, *p* 0.007). These features occur alongside a biomarker index for learning disorder (*n* 61, *rho* 0.347, *p* 0.006). Nearby are mixed methylation biomarkers of elevated vitamin B2, low folate, and low zinc with respect to copper, which eventually gives way to low methylation signatures, with elevated catecholamines implying restricted catecholamine metabolism. There is later development of several neuro-sensory processing disorders along with reactivated symptoms of auditory hallucinations, hostility, and delusions. In this disordered developmental setting, with mixed mood symptoms and biomarkers representing vacillating methylation states, it is not unexpected that *MTHFR* 677 CT carriers experienced the longest duration of illness (47 years), with a high Symptom Intensity Rating (SIR), (*n* 61, *rho* 0.808, *p* 0.000), frequent hospital admissions (*n* 61, *rho* 0.806, *p* 0.000), and low Social, Occupational (SOFAS), (*n* 58, *rho* −0.842, *p* 0.000), and Global function (GAF), (*n* 58, *rho* −0.844, *p* 0.000).

## 3. Discussion

### 3.1. General Considerations

In this study, we specifically focused on predictive biomarkers related to the duration of illness for each *MTHFR* 677 variant. In a graphed two-dimensional plane, we observed molecular biomarker correlates emerging and aligning with each other. Upon further observation, we noted how changes in allele combinations across graphs differentiated symptom clusters accompanied by phase changes in methylation states. Thus, psychosis symptom expression appears to be differentially regulated by *MTHFR* C677T variants and enzyme cofactors associated with methylation dynamics.

Research projects that map the progression of psychosis [91,92] are less prevalent than those that focus on risk prediction and risk of transition to psychosis scores [93,94]. This usually requires machine learning analysis of data that have been adopted from association studies or from large, prospective study cohorts [95,96]. Collecting data for such studies is a lengthy and expensive process. In these studies, the use of proteomic biomarker profiling for the prediction of psychosis has identified dysregulation of the complement and coagulation systems, which is thought to be related to an underlying immune or inflammatory response [97,98]. Our study takes another approach and provides a virtual snapshot of psychosis development derived from a single data collection for a small number of participants in a local catchment area. It ranks the correlates of predictive biomarkers along a virtual timeline of illness duration to construct a trajectory of biomarker relationships that can be understood and used for remedial purposes. With further research development, this approach may be useful for informing personalized patient care, in real-time, in a local context (Section 4.6). Because of the safety relevance of suicidality and hostility in acute clinical settings, we specifically discuss the relationship between predictive biomarkers for suicidality and hostility. On the virtual timelines described in Section 3.6 and Section 3.7, these symptoms emerge at times of change in methylation state, with illness phase transition, alongside a history of abuse, head injury, and other neuro-sensory loss. In this correlative context, we cannot make causal associations between these variables however we note that other authors report similar concerning associations [99,100,101].

### 3.2. Implications for Clinical Monitoring and Management

Within these timelines of illness progression for all *MTHFR* C677T variants, diagnostic certainty was only reached in the final phase of the illness progression, by which time frank features of psychosis had developed. Therefore, opportunities to correct abnormal biomarkers of illness progression had long been bypassed in the early phases of the illness. This means that there is ample time available for therapeutic interventions to prevent emerging psychosis and its adverse outcomes. An Adjunctive Model of Stepped Care Management (Figure 5) is therefore proposed, with the potential to assist the clinician to manage the biomarker features of psychosis development in a phase-informed manner. This plan seeks to establish and stabilize methylation states and their related abnormal biomarkers, to optimize neuronal transmission before later, remedial forms of treatment are initiated. In establishing biomarkers of methylation, it is necessary to recognize that frequent monitoring is necessary, because these biomarkers and their feedback mechanisms can adaptively adjust the methylation state in response to unfavorable initial conditions imposed by any particular genetic variant. This capacity to adjust the initial methylation state explains why a patient’s mental state may change as their psychosis develops and this may account for the perplexing variation in psychiatric diagnoses that a patient may receive over the duration of their illness. For this reason, molecular biomarkers that provide diagnostic and clinical certainty by longitudinal monitoring of the illness phases should be a welcome addition to psychiatry management.

The prevention or delay of illness progression or the development of adverse outcomes, such as suicidality, hostility, and disability, is a strategic aim of this project. During the acute onset of psychosis, pharmacotherapy is started. However, consideration may also be given to also commencing biomarker-informed biochemical treatment, with a gradual tapering of medication as biochemical interventions take effect, and symptom remission occurs. Implementing such adjunctive therapy requires a nuanced, integrative understanding of the biochemical pathways and expectations related to each *MTHFR* C677T gene variant, its methylation state, and potential adaptive methylation changes. This requires frequent biomarker monitoring and comprehensive knowledge of natural supplement options and their appropriate use within biochemical pathways, as well as dosing strategies, potential side effects, and possible interactions [102,103].

All molecular biomarkers emerging during the progression of illness in this study relate directly or indirectly to the ability of different *MTHFR* 677 variants to code for the MTHFR enzyme governing folate metabolism and thereby to control dynamics within the methylation cycle. These cycles are strategically coupled to fundamental biochemical processes that take place within mitochondria. These intracellular organelles reside within the body and brain, as well as within cells of the gastrointestinal wall and microbiome. The results of this study, therefore, highlight the nature of schizophrenia as a metabolic disorder and join a growing body of literature acknowledging that mitochondria are important sites for leveraging therapeutic advances in neurology and psychiatry [104]. It is also necessary to understand that the role of the genetic variant in determining the biochemical profiles observed during psychosis progression may not always align with expectations. As mentioned above, and shown in this study’s results, intrinsic biochemical mechanisms may internally regulate restrictions or excesses of intermediate substances, imposed by variable genetic effects on enzyme activity. At the same time, these regulatory mechanisms synchronize their activity to influence remote pathway enzymes engaged in serotonin and catecholamine metabolism or oxidative stress reduction.

While the symptoms of an acute psychosis episode necessitate immediate use of antipsychotic medication, the evaluation and treatment of underlying hidden needs in the form of stressors, risk factors, visual or auditory deficits, and neuro-sensory processing disorders, should not be forgotten. For patients in remission from psychosis, relaxation and stress-reduction techniques, along with cognitive-behavioral approaches, can significantly support the brain as it adjusts back to reality [105]. Therapeutic interventions such as family therapy, behavior therapy, and mindfulness therapy [106,107] are particularly valuable as they rely on effective communication between the therapist and the recovering individual. However, optimal outcomes from these therapies are only likely to be achieved after normalizing methylation and all biochemical and nutritional factors, after which sensory processing and stress reduction factors can further facilitate brain recovery. As progress is made in cell stabilization and regeneration, the stage is set to obtain optimal outcomes for treatment of auditory and visual processing disorders, as well as for cognitive rehabilitation and social and life skills training (Figure 5).

Sensory deficits in hearing and vision and dual visual and auditory sensory processing disorders seem common to all three *MTHFR* C677T variants and are, therefore, important considerations in the management of schizophrenia and schizoaffective disorders [108]. These disorders may arise from similar sources, such as a history of ear infection, hearing loss, head injury, or low methylation state with oxidative stress. These factors arise in the mid-illness trajectory of *MTHFR* 677 CC and CT variants, whilst auditory pathology on otoscopy relates to the late illness phase of the *MTHFR* 677 TT variant.

Two primary types of auditory processing difficulties have been identified. The first involves challenges in distinguishing a discrete auditory signal against ambient background noise or a visual “figure” against the “ground” noise of a dynamic visual background. These sensory discrimination difficulties may be linked to disruptions in the brain’s ability to switch attention between the central sensory-motor cortex, object-recognition, parvocellular pathways and the background-scanning magnocellular pathways in the outer occipital-parietal-temporal brain regions [109]. While “figure-ground” sensory discrimination issues may be genetically predisposed, a missed developmental window for auditory phonological coding during early language acquisition has been noted [110], and visual discrimination difficulty has been found related to dyslexia and learning disorders [111,112]. The second type of neuro-sensory processing difficulty involves the brain’s ability to integrate information between the left and right hemispheres—a function managed by the corpus callosum, which transfers auditory information from one side of the brain to the other [88]. Disrupted interhemispheric auditory communication in this region can lead to “competing words” difficulty, whereby patients struggle to discriminate between different words heard simultaneously in the left and right ears. This form of auditory processing deficit emerged as a significant biomarker in this study (Appendix A).

It is helpful if the staff in acute care wards learn to adjust their phenomenological insight to recognize patients displaying the subtle signs of visual and auditory sensory impairment. These signs are often embedded within the symptoms of psychosis and are part of its development. For instance, patients may exhibit verbal response delays that can be incorrectly interpreted as “thought-blocking”. Or, they may make repeated requests for verbal repetition. Alternatively, they may squint, avoid eye contact, or demonstrate visual uncertainty or distraction. Often, there is dual auditory and visual processing difficulty with profound attention deficit demonstrated by anxious, uncertain demeanour or cognitive attempt to boost attention by engaging in obsessional thought or self-stimulating motor movements. Clinicians who are knowledgeable about these defences learn to recognize them within the phenomenology of schizophrenia and schizoaffective disorders. Indeed, once acute symptoms have subsided, clinicians may exclude or confirm the presence of these conditions by conducting clinical assessments, using equipment such as that described in Appendix A. So prevalent are these sensory conditions within the schizophrenia condition, that such assessments should be conducted following every acute psychosis episode and follow-up ENT or ophthalmological evaluations should become a routine part of every psychosis discharge plan (Figure 5) [113].

Finding therapists to remediate sensory processing disorders is not always easy. Treatment options for such disorders now constitute a vast neurological field that contains many assessment and treatment applications. Within this field, behavioral audiologists, ophthalmologists, optometrists, and occupational therapists may adopt different treatment repertoires. Therapists employ various techniques, ranging from treatment for attention deficit disorder to biofeedback, orthoptic exercises, sound therapy, and occupational therapy. Sensory-motor and visuo-auditory integration protocols and brain stimulation treatments are also useful. Dichotic listening sound therapy is available on YouTube, with stereo earphones providing a therapeutic advantage [114].

### 3.3. MTHFR 677 CC Variant Specific Clinical Monitoring and Treatment

On linear regression analysis for DOI, the *MTHFR* 677 CC variant yielded a predictive model containing elevated catecholamine biomarkers that are characteristic of a low methylation state. As discussed in the Introduction, Section 1.3, Section 1.4, Section 1.5 and Section 1.6, these variables appear related to increased riboflavin metabolism with insufficient riboflavin available for flavin adenine dinucleotide (FAD) to support of the activity of the MTHFR enzyme coupled to reduced S-adenosyl methionine (SAMe) availability for COMT catecholamine metabolism, that has also been reduced by insufficient FAD to facilitate the monoamine oxidase B enzyme.

In this low methylation setting with conserved, elevated levels of catecholamines, ranked correlates for the *MTHFR* 677 CC variant identified five graphed phases of illness progression (Figure 2). In the early phases, low vitamin cofactors linked to low methylation occurred alongside attention impairment, auditory and visual processing deficits, and early dissociation symptoms. These were preceded by multiple risk factors, including a family history of mental illness, subclinical head injury, and history of ear infections, together with lowered auditory working memory and oxidative stress, followed by elevated histamine levels and intensified dissociative and obsessional symptoms. Many aspects of this low methylation state can be successfully treated by direct replacement of activated cofactor forms of folate vitamin D, B6, B2, and minerals [115] along with methyl donor molecules such as methyl folate, methionine, methyl cobalamin (methylated vitamin B12), or trimethyl glycine [116,117]. At the same time, it is important that regular blood and urine monitoring is undertaken to offset the risk of over-supplementation, which is expected to initiate low methylation feedback mechanisms. Although the direct methyl donor SAMe may play a future role in low methylation management [117], it is a somewhat unstable molecule and its direct effect on remote methylation pathways requires further research to offset any risk of proliferative disease or adverse effects arising from its powerful influence as a methyl donor in more remote biochemical pathways [118].

Probiotic organisms that synthesize various vitamins may also play a therapeutic role. For example, many *Bifidobacteria* species synthesize folate [119], and other microbiota such as *Bacillus subtilis*, *Lactobacillus fermentum*, *lactobacillus plantarum*, and *Escherichia coli* synthesize riboflavin [63,64,65,119,120,121]. However, probiotic use is challenged by durability issues within the gut, as vitamin absorption is reduced in achlorhydria [122], and other organisms in the gastrointestinal system may inhibit their efficacy. In addition, probiotics may require targeting to treat specific methylation states or tailoring to meet the challenge of adaptation to methylation shifts in both internal cellular and extracellular compartments. Targeted probiotic treatment trials that consider methylation variants and phenotypes are therefore urgently required. Consideration of existing medications is necessary prior to probiotic treatment, as oral contraceptives, proton pump inhibitors, nitrous oxide, ibuprofen, and some antibiotics, as well as metformin and aspirin, may inhibit methyltransferases or alter the microbiome in a manner that suppresses methylation [123,124]. There is also a need for the exclusion of pathology and chronic infection and inflammatory or immune challenges, which may perpetuate an over-methylation stress response that continues to deplete catecholamine reserves despite any underlying gene variation or other metabolic considerations [125].

Dietary issues have the obvious potential to influence vitamin and trace element availability, and low intake of either type of cofactor requires exclusion, particularly in the elderly, where folate and riboflavin deficiency may accompany declining self-care and dietary intake [126,127]. In such situations, riboflavin may be supplemented by the dietary intake of fresh meat, green vegetables, milk, and legumes, and baker’s or brewer’s yeast in bread and beer also provides high levels of riboflavin [128] as well as trace elements of zinc, magnesium, and potassium [115].

Diabetes type 2 was an early biomarker significantly predictive of the duration of illness for the *MTHFR* 677 CC variant in this study. Aged populations are particularly vulnerable to dietary riboflavin deficiency, and this condition has an established link with insulin resistance, diabetes type 2, and schizophrenia [129]. Riboflavin replacement has been shown to significantly reduce insulin resistance [130] and insulin resistance may be reinforced by the low activated vitamin D levels [131] found related to the CC variant. The additional contribution of the adrenal glucocorticoid response to insulin resistance in chronic stress conditions provides a further reason for stress reduction interventions [132]. Also, further considerations are required for replenishment of serotonin, as low vitamin B6 and vitamin B2 availability inhibits precursor tryptophan metabolism [29] and vitamin B6-dependent conversion to serotonin. Activated vitamin B6 is also required for the synthesis of glutathione in the trans-sulphuration pathway (Figure 1); therefore, well-monitored vitamin B6 replacement and/or glutathione replacement may help offset oxidative stress, in the low methylation state [133,134]. Biomarker correlates for DOI were also noted for elevated histamine levels in this variant, and associated symptoms of insomnia, daytime fatigue, amotivation [135,136,137], nasal congestion, or allergies, may be assisted by a low histamine diet or use of antihistamine medication.

The finding that frank psychosis symptoms emerge at the end of the timeline of the five illness phases of the MTHFR 677 CC variant suggests that there is considerable time available for therapeutic intervention before frank psychosis is reached. For instance, the history of ear infections as a factor presenting early in the illness trajectory is potentially remedial, and its thorough treatment may offset the development of auditory processing and learning disorder and perhaps prevent the later contribution of these disorders to psychosis symptoms [108]. Therefore, a preventative attempt to exclude and treat any chronic post-infective auditory deficits contributing to auditory processing delay [138] is mandatory to prevent psychosis developing in carriers of this CC variant. Education on ear hygiene, early treatment of ear infections, and later exclusion of auditory attention deficit or auditory processing disorders may be addressed in preventative educational programs. Likewise, early vision care education and visual acuity assessments to exclude subtle visual pathology should be followed by tests to exclude residual visual processing disorder.

### 3.4. MTHFR 677 TT Variant-Related Implications for Illness Monitoring and Treatment

As described in Section 2.7, and displayed in Figure 3, the homozygous *MTHFR* 677 TT variant displayed a distinct absence of abnormal biomarkers or adverse outcomes in the early course of illness, where there was optimistic mood and good neuro-sensory processing [139]. However, there was a sudden mid-illness trajectory shift in the biomarker pattern, preceded by suicidality and the onset of depressed mood with low motivation and social apathy. We speculate that such a shift may represent the onset of a depressed phase within bipolar affective illness, where elevated noradrenaline levels may reflect a low methylation dynamic. These mid-trajectory depressed symptoms contrast markedly with end-phase psychosis symptoms, which display symptoms of disorganization and hyperactivity alongside biomarker signatures of high methylation and activated psychotic symptoms, which may represent an overactive—possibly manic state.

As previously explained in the Introduction Section 1.3 [28,29,75,108], this metabolic shift may be related to two dynamics. Firstly, FAD is a flavin product of vitamin B2 (riboflavin), which is elevated in the MTHFR 677 TT setting [28]. FAD is unutilized by ineffective MTHFR enzyme coded by the *MTHFR* 677 TT polymorphism and it is possible that high vitamin B2 (riboflavin) levels may represent a similarly unutilized riboflavin dynamic. FAD is readily available for the metabolism of catecholamines by mono amine oxidase (MAO_B_). Secondly, over-compensative methylation of homocysteine by the zinc utilizing BHMT enzyme allows an alternative route for methionine to be methylated to form SAMe. SAMe facilitates catecholamine metabolism by facilitating the enzyme catechol-o-methyl transferase (COMT). These two effects (MAO_B_ and COMT) may combine to enhance catecholamine metabolism and deplete catecholamine levels in the highly methylated state. Indeed, despite evidence of DA sufficiency, we observed biomarkers of rapid catecholamine metabolic turnover this in the DOI-related correlative data for this variant, where the MHMA/NA + AD biomarker reflects a MAO_B_ effect, and the HVA/DA biomarker reflects a COMT effect. In the late phase of illness duration, these biomarkers occurred alongside anxiety and highly activated psychotic symptoms that may represent heightened cognitive defenses against depleted over-metabolized catecholamines.

However, we also noted the negative valence of the DA X 5-HIAA predictive biomarker, which was found together with markers for high vitamin B2 on linear regression analysis (Table 2). This biomarker implies that low dopamine levels may also predict extended DOI for the *MTHFR* 677 TT variant [89]. Although many processes may impact the level of the urinary serotonin metabolite 5-HIAA, vitamin B2 metabolizes serotonin’s precursor molecule, L-tryptophan, in the kynurenine pathway [140]. Therefore, low 5-HIAA may reflect low L-tryptophan availability for serotonin synthesis, further explaining the mid-phase DOI-correlated symptoms of low motivation, social apathy, and marginal suicide risk found in the mi-trajectory of the *MTHFR* 677 TT variant. We consider this to be evidence that there is a risk of both dopamine and serotonin instability, with dopamine available, but being excessively metabolized (HVA/DA) in this TT variant. In this regard, we note reports symptoms of schizophrenia with phases of low dopamine levels [141]. We also noted a link between the homozygous MTHFR 677 mutation and bipolar affective disorder [27], while other researchers have reported this variant’s association with low dopamine in Parkinson’s disease [142]. Therefore, if renal function is normal and the biomarker of excessive dopamine metabolism (HVA/DA) is detected on urine monitoring, consideration should be given to minimizing dopamine receptor-blocking antipsychotic medication. Another consideration is to modify high methylation-related excessive catecholamine metabolism by restoring methyl folate (5-MTHF) to offset the need for compensative over-methylation. Since this study shows that high B6 levels predict DOI for the *MTHFR* 677 TT variant (Section 2.4), one further consideration is that monitoring vitamin B6 levels and avoiding vitamin B6 supplementation may prove to be important for *MTHFR* 677 TT carriers, in order to reduce risk of the peripheral sensory neuropathy syndrome that accompanies B6 hypervitaminosis [143].

Since SAMe adds methyl groups to histones and cytosine DNA bases, future management of a high methylation phases may extend to using histone deacetylase inhibitors (HDAC inhibitors) or DNA methyltransferase inhibitors (DNMT inhibitors) [144,145]. These have been trialed in other pathological settings [146]. Small interfering RNA (siRNA) may also be considered to recruit DNA methyltransferases prior to their inhibition, resulting in methylation suppression by loss of proteins in methyl-DNA binding sites [147].

### 3.5. MTHFR 677 CT Variant-Related Implications for Illness Monitoring and Treatment

The heterozygous *MTHFR* 677 CT variant (Figure 4) achieved predictive strength for mixed biomarkers of high and low methylation. This mixture gave a biomarker trajectory, with an early high methylation phase, followed by a later swing to a low methylation state and some mixed characteristics. As the duration of the illness progressed, the biomarkers gradually gave way to increasing numbers of low-methylation signatures with elevated catecholamines, implying restricted catecholamine metabolism due to riboflavin (FAD) or SAMe insufficiency (Figure 4). Such instability was accompanied by a mixed field of symptoms that may represent a mixed or rapid cycling affective state. In this field, early dissociated, obsessional, depressed, manic, disorganized, and reorganizing symptoms gave way to several neuro-sensory processing disorders as frank psychosis symptoms took hold. By this time, mounting numbers of abnormal biomarkers had passed, suggesting that there was considerable time available for therapeutic intervention to prevent progression to the final phase of frank psychosis. For instance, the history of ear infections presenting early in the illness trajectory represents a preventative opportunity where follow-up auditory assessments may offset later development of auditory processing and learning disorder [108]. In this context, it requires noting that the history of abuse aligned with bone conduction abnormality (indicative of hearing loss) and subclinical head injury are collateral timeline factors that may set the scene for developmental delay with learning disorder and early onset suicidality [101].

Due to the metabolic and symptomatic instability of this *MTHFR* 677 CT variant, close biochemical monitoring of methylation status is advisable, but may yield contradictory results. Consideration of the use of mood-stabilizing medications may be an appropriate treatment option, although it is noted that valproate may interfere with downstream DNA methylation [148]. Another option is the use of natural methylation-stabilizing adaptogens, such as curcumin, quercetin, sulforaphane, and betaine. These adaptogens are found in a nutrient-dense diet but may be boosted by supplementation, with methylation monitoring [149]. When there is a concurrent elevation of vitamin B2 and low zinc with respect to copper, the high methylation state may settle with 5-MTHF (methyl folate) supplementation to suppress the need for compensative activation of alternate methylating pathways. Methyl folate is also not unwelcome in a low methylation state, if there is regular monitoring and low vitamin levels are also corrected. The molecular biomarker vitamin B12/activated vitamin D proved to be a significant predictor of DOI in the *MTHFR* 677 CT variant. Since Vitamin B12 is a required cofactor for homocysteine conversion to methionine in the methylation cycle, vitamin B12 may be unutilized when methylation is constrained by limited activity of the MTHFR enzyme or if methylation is diverted into the compensative BHMT pathway.

### 3.6. Suicide Prevention and Phase of Illness Monitoring for All MTHFR C677T Variants

Collectively, these biomarker findings suggest that a shift in methylation status—from low to high (CC), high to low (CT), or into a catecholamine-depleted state (TT), may be related to mood instability and risk of suicidality. For this reason, early assessment and regular metabolic monitoring to detect methylation changes are expected to assist in suicide prevention. The exclusion of related pathological conditions and auditory and visual neuro-sensory defects as part of the proposed stepped care management (Figure 5) will also play an important role in suicide prevention.

### 3.7. The Meaning of Hostility Across MTHFR C677T Variants

While the emergence of hostility has correlative strength in mid-illness phases for both *MTHFR* 677 CC and CT variants, hostility is highly correlated with DOI in the late phase for the *MTHFR* 677 TT variant (Section 2.7). In the diagnostic setting of this variant, hostility is related to manic-flavored psychosis symptoms and high methylation biomarkers. In this setting, we also noted a correlation for reduced visual span, which may be related to elevated adrenaline (AD/NA ratio), which is attributed to the plentiful availability of the methyl donor SAMe for facilitating the conversion of NA to AD. High levels of adrenaline are associated with the fear response and pupil dilation, which raises intra-ocular pressure, causing blurred vision with reduced visual contrast sensitivity and impaired visual discrimination. This may contribute to late-phase symptoms of anxiety, suspiciousness, and hostility as visual clouding activates a stress response [85,114]. Where anxious agitation or paranoid fear is a cause for clinical concern, there may therefore be a role for beta-blocker medication to modify receptor response to high adrenaline levels. From the clinician’s perspective, such behavior gives the impression of an increased intensity of illness (SIR), which is a strong biomarker outcome for the *MTHFR* 677 TT variant.

### 3.8. Study Limitations and Strengths

We acknowledge that our novel approach of rank-ordering correlate strengths of predictive biomarkers based on illness duration has certain limitations as correlates serve as indicators of relationships and are not direct measures of causality. Despite its advantages in displaying biomarker interactions over the course of illness, the timeline axis onto which we mapped duration-correlated biomarkers was a virtual construction rather than an absolute representation.

The strength of our biomarker findings can be attributed to the careful selection of candidate markers, recruitment of well-defined symptomatic participants, and efforts to minimize confounding variables. The meaning of the biomarkers and their methylation states was deduced from maps of established biochemical pathways and existing literature on methylation dynamics [30]. As detailed in the Methods Section 4, our final analysis utilized paired rather than matched data—a legitimate approach that can be more efficient, as matched analyses may inadvertently match risk factors when mental illness disorders share common risk profiles [150]. A key limitation was the small sample size of lower-prevalence MTHFR 677 TT carriers, which impacted the analysis of the final variant sample. Nevertheless, strong biomarker signals were identified, and the sample size for the TT variant aligned with the global prevalence rates and met the lower range for the Australian population [34]. The borderline statistical power for this TT sample suggests that the observed significant results are likely to be accurate; however, some true associations may have been missed. Some potentially confounding variables could not be controlled, such as smoking habits and the use of non-excluded medications by some participants. For example, oral contraceptive use can affect the levels of vitamin B6, folate, and vitamin B12 [151], while sodium valproate has been shown to reduce folic acid levels, increase homocysteine, and act as a histone deacetylase inhibitor [152]. Additionally, thyroid hormone levels, which were not measured in this study, could influence flavin mononucleotide (FMN) synthesis from riboflavin [153].

Although there is growing recognition that mental illness represents a systemic metabolic disease [154], we acknowledge that the observed associations between methylation patterns and vitamin levels across *MTHFR* C677T variants may not necessarily translate to specific tissue sites. Furthermore, the Peak 1 riboflavin-related analyte observed in the HPLC analysis is presumed to be a metabolite fraction but remains unidentified—this will be addressed in future research.

This research requires further development. Therefore, all opinions and implications for treatment expressed in this manuscript are considerations that are subject to the caveat that the study results have been replicated and validated. To achieve validation and build on these findings, it would be valuable to assess a broader array of genes associated with the biomarker landscape, DNA expression levels, and additional neuroinflammatory markers. Measuring N-methyl-D-aspartate (NMDA) receptor function could also provide insights, as the tricarboxylic acid (TCA) cycle, which is supplied by pyruvate from glycolysis, requires riboflavin-activated vitamins. Reduced vitamin B6 activation may inhibit glutamate output from this cycle, contributing to the NMDA receptor hypofunction commonly reported in schizophrenia [155]. A panoply of other key genetic variants influencing the activity of enzymes in the folate, methylation, and catecholamine pathways also awaits further research. For instance, the polygenic predictive power of betaine homocysteine methyl transferase gene variants, monoamine oxidases, and catechol-o-methyltransferase (*COMT*) gene variants alongside methionine adenosine transferase (*MAT*) and S-homocysteine methyl transferase (*SHMT*) gene variants has yet to be investigated. Furthermore, there is ample potential for these methylation-related gene variants to influence downstream DNA methylation, thus affecting epigenetic gene expression.

## 4. Materials and Methods

### 4.1. Summary of Study Design

Patients diagnosed with schizophrenia or schizoaffective disorder and participant controls from the same clinical catchment area (*n* 134) had *MTHFR* C677T variants identified in plasma from participant blood samples (Section 4.4, Appendix A). Promising molecular, and neuro-sensory biomarkers and risk factors predictive of diagnosis for each of the three MTHFR C677T variants (CC, TT, and CT) were identified by Receiver Operating Characteristics (ROC) analysis (Section 4.5, Appendix A).

The pre-identified diagnostic biomarker variables were subjected to linear regression analysis with respect to the patient-reported duration of illness (DOI) in years to discover which biomarkers were able to predict DOI (Section 4.5, Appendix A).

Biomarkers identified using the above methods were then analyzed using Spearman’s correlation analysis to determine their strength in relation to the DOI for each MTHFR C677T variant (Appendix A). Biomarker correlation strengths for significant (*p* < 0.05) risk factors, biochemical, and neuro-sensory biomarkers were ranked from least to greatest strength (Appendix A), and strengths plotted according to their biomarker category around a four-sided graph matrix constructed for each of the three MTHFR C677T variants (Figure 2, Figure 3 and Figure 4). Biomarker categories were linked by a third equidistant oblique axis, which served as a virtual timeline along which biomarker correlate strengths were projected and observed to emerge with respect to each other and with illness symptoms and outcomes (Figure 2, Figure 3, Figure 4 and Figure 6).

### 4.2. Recruitment

This study was conducted in the western area of Adelaide, South Australia, under the auspices of Queen Elizabeth Hospital, Woodville. Ethical permission for this study was provided by the affiliated Research Ethics Committee. All participants were informed of the study aims and methods and provided written consent to participate and have results published. All participants were rated for subclinical symptoms, and a family history of mental illness was not a recruitment exclusion factor (Appendix A). Further details of the inclusion and exclusion criteria are presented in Appendix A. The rating measures are described in Appendix A, the assessment techniques are outlined in Appendix A and the Case Control recruitment data are presented in Appendix A.

### 4.3. Data Collection

Data were collected from 134 multiethnic participants in the age range of 18 to 60 years. Multiple exclusion criteria were imposed on all study participants to strip the functional form of psychosis down to its essential condition. This meant excluding participants who were substance abusers or who had been investigated for organic causes of psychosis. Participants with any form of motor disability, extra-pyramidal symptoms of note or perceived or documented neuro-sensory loss, or any neurophysiological condition capable of confounding outcome measures on visual or auditory assessments were also excluded. Medications that were similarly excluded were those capable of confounding candidate marker results (Appendix A) and those that remained stable throughout the assessment window. In this manner, only highly characterized cases of functional psychosis were recruited.

### 4.4. Assessment Methods

Data were collected for demographic biomedical, physiological, symptomatic, and adverse functional outcome variables. Participants’ case notes and a standard interview protocol were used to collect demographic data related to the history of development, organicity, biochemistry, and neuro-sensory-processing disorder. These records included age, reported duration of illness [DOI] in years, history of developmental delay or disorder or learning disorder, subclinical head injury, past ear infection, accessory psychiatric diagnoses, and medical comorbidity. Hospitalization frequency, disability support requirements, medical comorbidities, and history of abuse of any type were also recorded. DSM-IV diagnoses for schizophrenia or schizoaffective disorder were checked by the DSM-IV R diagnostic checklist, and all participants were rated on standardized outcome scales, including the Brief Psychiatric Rating Scale [BPRS] combined with the Positive and Negative Syndrome Scale for Schizophrenia [PANSS], where each symptom was rated on a scale from 1 to 7 for intensity. These ratings summated to form a Symptom Intensity Rating [SIR] index, which provided a surrogate measure of symptomatic (clinical) severity. Outcome rating measures were the Social and Occupational Functioning Assessment Scale [SOFAS], Global Assessment of Function [GAF], and Clinical Global Impression rating [CGI] (Appendix A). Our data included those from undiagnosed participants from the same catchment area, who acted as controls for diagnostic ROC analysis purposes. 25% of these participants had a family history of mental illness, and these individuals were evaluated in the same manner as patients, with assessment of subthreshold symptoms, risk factors, neuro-sensory processing variables, and adverse outcomes.

As noted above, the *MTHFR* 677 gene is known to contain a single nucleotide polymorphism (SNP), and blood samples were collected for plasma detection of the wild-type (CC) sequence, homozygous (TT) point mutation, and heterozygote (CT) DNA with wild-type and mutant strands. These variants were identified in a licensed commercial laboratory using a Roche Diagnostics Light-Cycler 480 kit obtainable from Roche Diagnostics Australia Pty Ltd., North Ryde, NSW 2113, Australia. This kit contains a FastStart DNA Master HybProbe that probes genotype detection using a melting curve analysis method to identify different variants. The probe melts off a perfectly matched sequence and a mismatched sequence at different melting temperatures. Further information and links to the assay kit’s commercial website are available in the second table of analytical methods in Appendix A. All other analytes were assayed using commercial means in licensed laboratories. These were urine catecholamines, DA, NA, and AD, 5-hydroxy-indole acetic acid (5-HIAA), intermediate substances homocysteine, histamine, and creatinine, and the activated forms of folic acid (5-MTHF) and vitamin cofactors D, B12, and B6. Riboflavin (vitamin B2) and its expected metabolite (Peak 2 and Peak 1, respectively) were determined by high-pressure liquid chromatography (HPLC) (Appendix A). Hydroxypyrroline-2-one (HPL), was assayed in urine as a marker of oxidative stress [28,30].

All details of the neuro-sensory processing, neurocognitive, and visual and auditory test procedures have been licensed and presented in previous research reports [28] and in Appendix A, with data characteristics presented in Appendix A and expanded information links provided in Appendix A. Visual assessment included tests of near and distance vision acuity test, visual (symbol) span test (as a measure of visuo-spatial working memory capacity), and threshold speed of visual processing performance test (VSOP %) with results expressed relative to norms for age, as a measure of the speed of visual processing. Although participants with gross neuro-sensory or conductive hearing loss were excluded from the study, subclinical differences between air and bone conduction were noted on an audiogram, and an otoscopy was conducted to exclude obvious ear pathology. Further auditory assessment included forward and reverse digit span tests of auditory attention and working memory capacity, threshold speed of auditory processing as a percentage of age and competing words differentiation measure for age as an indicator of intra-cerebral dichotic listening performance.

### 4.5. Data Analysis

A description of the study design with respect to the data analysis methodology is provided in Section 4.1. The participant recruitment processes, and original data preparation and pairing are described via links in Appendix A. Our final analysis used paired, rather than matched, data. This approach is legitimate and can be more efficient, as matched analyses might inadvertently match risk factors when mental illness disorders share common risk profiles [150]. *MTHFR* C677T polymorphism distribution between the case and control groups was confirmed by crosstab investigation (Appendix A). The overall data set was then split into three according to the three *MTHFR* C677T variants, after which the mean sample characteristics for the reported DOI were determined using STATA, Release 17 software [156] (Appendix A). This software was used to perform Receiver Operating Characteristics (ROC) analysis on the three *MTHFR* C677T variant data sets to determine which variables were able to significantly discriminate between the case and control populations (Appendix A). ROC analysis provides the area under the ROC curve (AUC). An AUC of 0.5–0.7 represents poor biomarker diagnostic discrimination, an AUC of 0.7–0.8 indicates acceptable biomarker diagnostic discrimination, an AUC of 0.8–0.9 indicates excellent biomarker diagnostic discrimination and an AUC of >0.9 indicates outstanding biomarker diagnostic discrimination. Diagnostic Specificity and Sensitivity are acceptable at ≥85 per cent and optimal at ≥90 percent. In addition to providing Area under the ROC curve (AUC) and significance levels (P), ROC analysis provides percentage parameters for Odds Ratio and Sensitivity and Specificity, along with Positive Predictive Value (PPV) and Negative Predictive Value (NPV). An Odds Ratio (OR) ≥ 2 is a notable finding, and Positive Predictive Value (PPV) and Negative Predictive Value (NPV) parameters greater than 85% are also considered important.

Significant variables with predictive diagnostic value were selected for further linear regression analysis in relationship to illness duration [157,158,159]. This analysis was conducted for each MTHFR C677T variant, with patient-reported duration of illness (DOI) in years as a continuous dependent variable. Several DOI predictive models were obtained for one of the MTHFR C677T variants containing clusters of variables (Appendix A). Nonparametric Spearman’s correlation analysis for DOI was then conducted for the respective MTHFR C677T variants on all predictive biomarkers found on diagnostic ROC analysis and linear regression analysis (Appendix A). (In these analyses, non-cases rated for subclinical symptoms were allocated a DOI of 0 (Appendix A). These analyses used SPPS v 25 software [160], and all tests were 2-tailed, and a *p*-value of <0.05 was considered significant. For continuous variables, the nonparametric Spearman correlation may be high and positive [or low and negative]. The same rule applies to a dichotomous [ROC] predictor, where DOI is greater as the strength of the correlate increases [or DOI is less as the strength of the correlate weakens]. The Benjamini-Hochberg procedure [161] was utilized to correct ROC, Odds Ratio, and Spearman’s correlation results for the effect of multiple comparisons.

### 4.6. Graph Matrix Construction for Ranked Correlates

To simulate the trajectory of symptom progression for each *MTHFR* C677T variant, significant (*p* < 0.05) DOI related biomarkers identified from ROC and linear regression analyses had their correlates ranked from lesser to greater strength (Appendix A). These ranked correlates were divided into risk factor, symptoms, biomedical marker, functional outcome, and neuro-sensory processing categories. Correlate strengths *rho* (RHO) for high and low biomarkers were plotted according to their strength along the outer edges of the x- and y-axes of a four-quadrant graph matrix. For instance, where there was a negative correlation for vitamin B6 (as was found for the CC variant), this was reinterpreted and mapped according to the positive correlation strength for low vitamin B6. The theory behind this method is that if one variable has a significantly higher, positive DOI correlate-strength than another, then that variable relates to a longer duration of illness. All graphed biomarker correlates were then projected onto a third equidistant oblique axis, which served as a virtual timeline. In this manner, we observed the progression of psychosis illness according to different significant risk factors, biomarker, neuro-sensory, and illness outcome variables. In this landscape, correlates with lower strength appear earlier on the virtual timeline, and correlates of larger strength occur later. The graphs constructed for biomarker correlated variables associated with DOI for each *MTHFR* C677T variant (TT, CC, and CT) were then compared to observe differences in trajectory characteristics among carriers of the three different variants. In this manner, biomarkers were visually appraised in relation to each other and to the progression of the illness.

## 5. Conclusions

The *MTHFR* C677T gene variant plays a crucial role in methylation biochemistry, and polymorphisms in this variant are linked to neurodevelopmental disorders and schizophrenia. By differentiating the relationship between predictive biomarkers for schizophrenia and schizoaffective disorder with the duration of illness for each *MTHFR* C677T variant and plotting ranked correlative data against the timeline of illness duration, biomarker emergence can be visualized along psychosis development trajectories. In these trajectories, risk factors, neuro-sensory deficits, and illness outcome biomarkers can be seen to consolidate into different phases of illness. These phases are related to changes in methylation states and alterations in enzyme cofactor availability in methylation, indole-catecholamine, and oxidative stress management pathways.

The detailed results and insights into the development of psychosis presented in this study begin to position *MTHFR* C677T gene variants and their methylation-related enzyme cofactors as meaningful contributors to symptom expression in schizophrenia and schizoaffective disorders. They allow for an enhanced understanding of the psychosis experience and its phenomenology. They have the potential for real-time monitoring in local settings and therapeutic applications to prevent or delay psychosis progression, or overcome treatment resistance. Major steps in psychiatric management may be achieved through an adjunctive stepped care management plan that is informed by the biomarker illness phases. Once neuronal function has been restored by biochemical adjustment, risk factors and neuro-sensory remediation can achieve more durable results, with improved therapeutic outcomes for persons with schizophrenia or schizoaffective disorder.

## Figures and Tables

**Figure 1 ijms-25-13348-f001:**
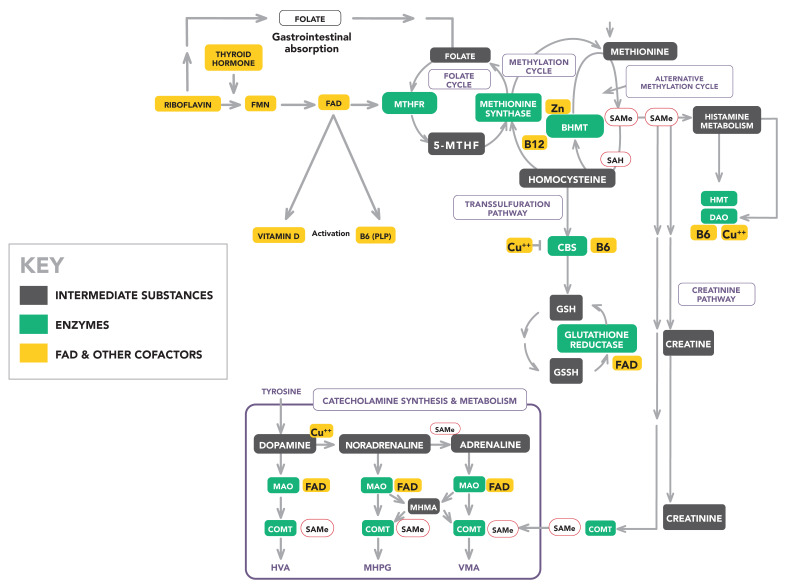
Methylation-related biochemical pathways. INTERMEDIATE SUBSTANCES: SAMe—S-adenosylmethionine, SAH—S-adenosylhomocysteine, MHMA—3-methoxy-4-hydroxymandelic acid, HVA—homo vanillic acid, MHPG—4-hydroxy-3-methoxyphenylglycol, VMA—Vanillylmandelic acid, GSH—Glutathione (reduced, active antioxidant form), GSSH—Glutathione (oxidized form). ENZYMES: MTHFR—Methylenetetrahydrofolate reductase, BHMT—Betaine homocysteine methyltransferase, CBS—Cystathionine Beta Synthetase, MAO—monoamine oxidase, COMT—catechol-o-methyltransferase, HMT—histamine methyl transferase, DAO—diamine oxidase. RIBOFLAVIN DERIVED COFACTORS: FMN—flavin mononucleotide and FAD—flavin adenine dinucleotide, OTHER COFACTORS: 5-MTHF—5-methylenetetrahydrofolate, B6 (PLP)—pyridoxal phosphate (the activated form of vitamin B6), B12—vitamin B12 (cobalamin). Cu^++^—Free (unbound) copper (Cu), Zn—zinc.

**Figure 2 ijms-25-13348-f002:**
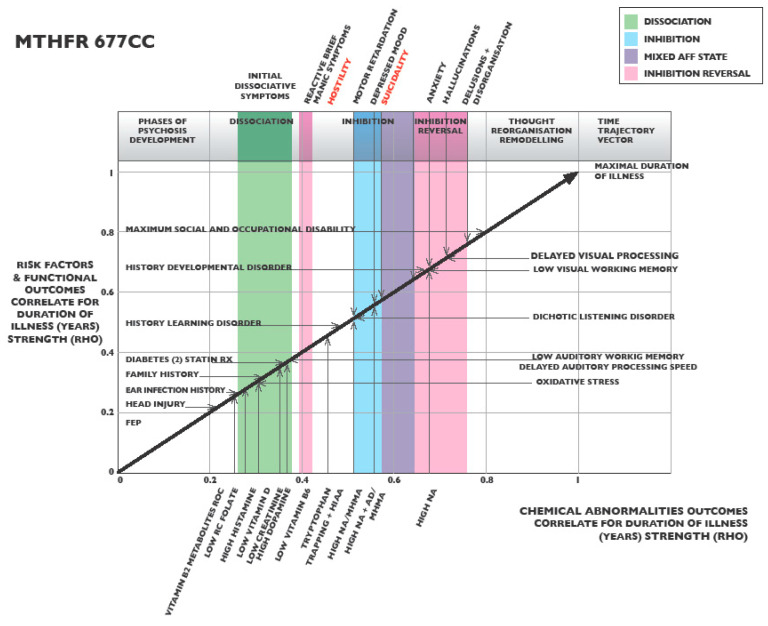
Mapping *MTHFR* 677 CC variant-related predictive biomarkers correlates with the duration of illness, where Spearman’s correlation *rho* (RHO) score range on the x- and y-axes = 0–1 and the diagonal black arrow is a virtual timeline trajectory. (Vitamin B2 metabolites ROC = Significant Receiver Operating Characteristic (Area under curve—AUC *p* < 0.05), for urine HPLC Peak 1 riboflavin co-analyte (presumed metabolite)).

**Figure 3 ijms-25-13348-f003:**
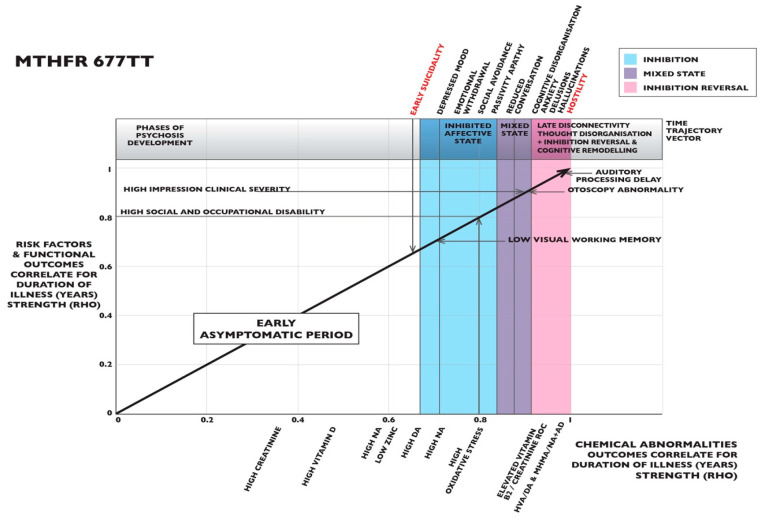
Mapping *MTHFR* 677 TT variant-related predictive biomarkers correlates with the duration of illness, where Spearman’s correlation *rho* (RHO) score range on the x- and y-axes = 0–1 and the diagonal black arrow is a virtual timeline trajectory.

**Figure 4 ijms-25-13348-f004:**
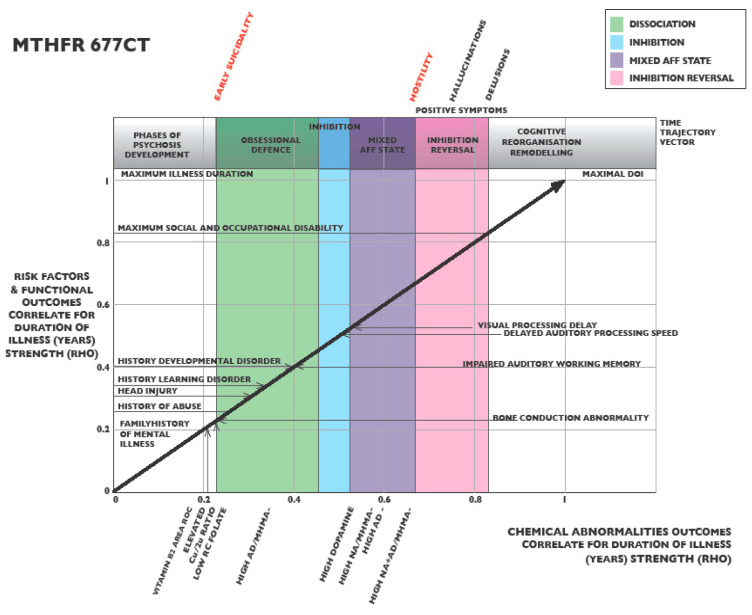
Mapping *MTHFR* 677 CT variant-related predictive biomarkers correlates with the duration of illness, where Spearman’s correlation *rho* (RHO) score range on the x- and y-axes = 0–1 and the diagonal black arrow is a virtual timeline trajectory.

**Figure 5 ijms-25-13348-f005:**
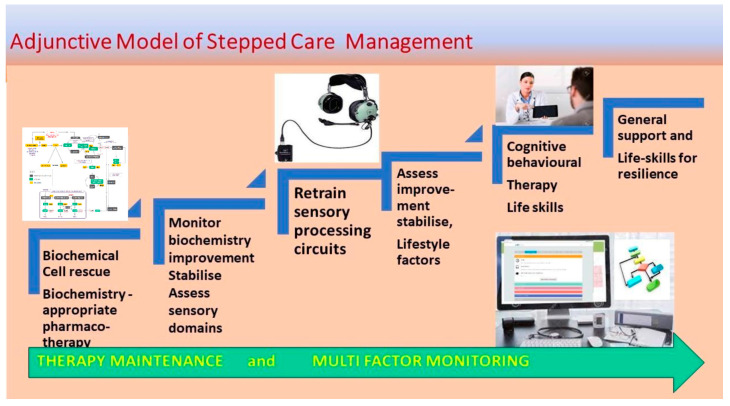
A stepped care approach to rehabilitation for psychosis.

**Figure 6 ijms-25-13348-f006:**
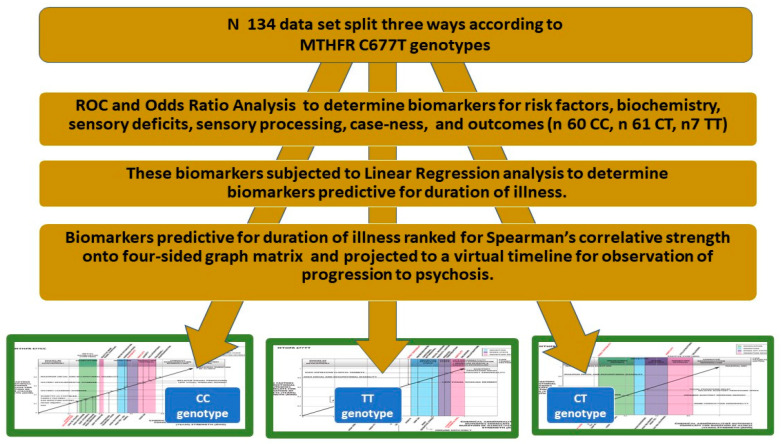
Method sequences for mapping the progression of psychosis.

**Table 1 ijms-25-13348-t001:** Linear regression analysis results for duration of Illness for *MTHFR* 677 CC variant dataset and case-related dataset.

DOI and *MTHFR* 677 CC	*n*	Variable	*b*	*p*
Biomedical variables		NA	0.320	0.000
alone		DA	−0.047	0.040
	58	AD/MHMA	1.267	0.002
		DA × 5-HIAA	0.003	0.006
Neuro-sensory processing variables alone		Visual span	−3.388	0.000
	52	Distance vision R	0.300	0.038
		ASOP % age diff	0.095	0.019
All biomedical, neuro-sensory	50	NA	0.131	0.027
and comorbid variables		Visual span	−0.950	0.152
Combined		HPL/creatinine	0.468	0.004
		Diabetes 2	15.175	0.000
		VSOP %	0.087	0.023
		AD	0.772	0.004
		hypertension	−5.984	0.014

*n =* number of variables, *b* = regression coefficient, *p* = significance probability (*p* < 0.05). (blue = low methylation biomarker, red = high methylation marker). NA = noradrenaline, DA = dopamine, AD = adrenaline, 5-HIAA = 5 hydroxy indole acetic acid, MHMA—3-methoxy-4-hydroxymandelic acid, HPL = Hydroxyhaemopyrroline-2-one, ASOP = Threshold speed of auditory processing, VSOP = Threshold speed of visual processing, distance vision R = Right eye distance vision acuity. (Legend of further abbreviations at beginning of Appendix A and in Appendix A).

**Table 2 ijms-25-13348-t002:** Linear regression analysis results for DOI, for full *MTHFR* 677 TT variant data set.

DOI and *MTHFR* 677 TT	*n*	Variable	*b*	*p*
Biomedical	6	Vitamin B6	0.007	0.041
variables alone		Vitamin B2/creatinine	48.318	0.009
		DA X 5-HIAA	−0.045	0.006

b = linear regression coefficient, *p* = significance probability (*p* < 0.05). Red text = high methylation state-related biomarkers. DA = dopamine, 5-HIAA = 5 hydroxy indole acetic acid.

**Table 3 ijms-25-13348-t003:** Linear regression analysis results for DOI for the *MTHFR* 677 CT variant.

DOI and *MTHFR* 677 CT	*n*	Variable	*b*	*p*
Biomedical variables alone	59	NA Histamine Vitamin D Vitamin B12/ vitamin D	0.27 9.12 0.19 0.51	0.000 0.038 0.032 0.018
Neuro-sensory processing variables alone	60	Reverse digit span Visual span Distance vision R	−2.09 −4.22 0.41	0.061 0.000 0.058
Biomedical, neuro-sensory, and comorbid variables combined.	47	DA × 5-HIAA NA NA/MHMA Vitamin D Vitamin B12/vitamin D Vitamin B6 Plasma homocysteine Histamine + NA Distance vision R BMI Dyslipidemia	0.005 −16.98 −0.007 0.34 0.82 −0.06 −1.68 17.24 0.68 0.73 −8.84	0.049 0.000 0.031 0.000 0.001 0.000 0.010 0.000 0.003 0.000 0.006

*n =* number of variables, *b* = linear regression coefficient, *p* = significance probability (*p* < 0.05). Red text = high methylation state-related biomarkers, Blue text = low methylation state-related biomarkers. DA = dopamine, 5-HIAA = 5 hydroxy indole acetic acid, NA = noradrenaline, MHMA—3-methoxy-4-hydroxymandelic acid, distance vision R = Right eye distance vision acuity, BMI = body mass index.

## Data Availability

Limited anonymous data are available from the corresponding author upon reasonable request.

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
