# Peer review of "Gene Variant Related Neurological and Molecular Biomarkers Predict Psychosis Progression, with Potential for Monitoring and Prevention"

_ijms, 2024, doi:10.3390/ijms252413348_

Round 1
Reviewer 1 Report
Comments and Suggestions for Authors
The manuscript entitled "Mapping neurological and biochemical biomarker correlates for schizophrenia, schizoaffective disorder and duration of illness differentiates discrete phases of illness dependent upon MTHFR C677T variant, with potential for phase-monitored therapeutic interventions to prevent psychosis progression and adverse outcomes." authored by Fryar-Williams et al. presented an analysis of the correlates that establish between different types of biomarkers of schizophrenia.
There are several major concerns:
- The title needs revision as it is extremely wordy; it should be short and representative;
- Abbreviations use is not consistent throughout the manuscript;
- "On linear regression analysis, molecular and neuro-sensory biomarkers found highly predictive of illness duration, were reanalysed by Spearman’s correlation" - Spearman's correlation shows a monotonic correlation, not necessarily linear;
- The use of punctuation should be revised; spelling and phrasing should also be revised;
- The paragraph on line 93 seems not to fit within the scope of a usual introduction;
- Figure 1: all abbreviations should be described in the figure legend;
- Suppl. S4 - Percentage Free Copper/Red Cell Zinc - if there's a previous description of this parameter, citation should be given;
- Suppl. S5 & 6 - How were these tests available? If they are licensed, this should be mentioned;
- line 168 - the writing of Latin names for the microorganisms has several rules;
- Section 5.1 - the Authors should briefly present the research demarch within this section, rather than presenting an extensive supplemental material;
- Results: the Authors should revise this section; rather than presenting a summary of the methods here, the Patients and methods section should be revised so that the summary won't be necessary here; also, maybe the Authors should concentrate Their attention more on what's significant and revise the Results section after presenting a more clear picture of the experimental design;
- the Summary section and all the interpretations of the results should be concentrated in a section representing the Discussions.
There are also minor errors and typos (eg. line 702 - hydroxyhemopyrroline-2-one, and others). Line 959 - This table seems misplaced and redundant (also presented in the Supplementary material).
Comments on the Quality of English LanguageI am not native to English, but the manuscript should benefit from revisions, as I found it hard to understand.
Reviewer 2 Report
Comments and Suggestions for Authors
Summary:
The authors present a compelling manuscript that offers a detailed examination of the various clinical symptoms associated with the MTHFR C677T variant. However, the methodology for variant identification needs to be clarified and expanded upon. Additionally, the Discussion section requires reorganization. This manuscript holds significant value due to its comprehensive overview of therapeutic interventions.
Title:
The title should use capital letters for each word, in accordance with MDPI’s Instructions for Authors. Additionally, I recommend shortening the title to improve visibility. Currently, the title is too long; using a column “:” to highlight key aspects like "therapeutic interventions and adverse outcomes" could be beneficial.
Abstract:
Lines 27-30: The sentence is too long and should be split into two parts. Also, it would be helpful to highlight the novelty of your study. While the MTHFR C677T variant is well-described, your work appears to offer new insights into therapeutic interventions. Make sure the last sentence in the abstract, which serves as the "take-home message," emphasizes this point.
How was the variant identified? Please specify the diagnostic method used, as this is currently unclear.
Keywords:
To increase visibility, incorporate specific terms that are not mentioned in the title as part of the keywords.
Introduction:
This section provides a thorough description of the conditions linked to MTHFR C677T. To improve accuracy:
Include the MIM phenotype codes associated with MTHFR to provide clearer information on the relevant conditions.
Discuss the inheritance patterns associated with MTHFR variants, particularly noting that it can exhibit autosomal recessive inheritance in certain conditions (e.g., neural tube defects).
Lines 82-103: While the study design is well-detailed, I recommend moving this information to the beginning of the Discussion section. In the last paragraph of the Introduction, briefly summarize the aim of your study in two sentences.
Results:
Separate the Results from the Discussion to improve the flow and readability. This will make it easier for readers to place your findings in the broader scientific context of MTHFR and the C677T variant.
Table 1: Ensure the table is formatted according to MDPI’s guidelines, and spell out all acronyms in the caption. Move the footnote explaining “b = Regression coefficient, p = significance probability” to the table footnote. Clarify the p-value range to show the significance of your results.
Figures 3 and 4: Add more detail to the figure descriptions. Specify the score ranges used on the x and y axes (e.g., 0-1), and explain the meaning of the diagonal black arrow (virtual 278 timeline trajectory).
Table 2: Apply the same formatting suggestions made for Table 1.
Figure 5: Improve the figure by adding PNG images of the chemical structures of the molecules mentioned.
Line 560: Correct “MTHFR$” to “MTHFR.”
Discussion:
Sections 3 and 4 should be reorganized as subsections within the Discussion. Additionally, you should create a dedicated Discussion section that reflects on your results in the context of existing literature and comments on the limitations of your study.
Materials and Methods:
This section is missing essential details regarding the technical approach used for variant identification. Please elaborate on the methodology and include relevant technical details.
Round 2
Reviewer 1 Report
Comments and Suggestions for Authors The quality of manuscript ijms-3262037, entitled "Gene Variant related Neurological and Molecular Biomarkers Predict Psychosis Progression, with Potential for Monitoring and Prevention," authored by Stephanie Fryar-Williams, Graeme Tucker, Peter Clements, and Jorg Strobel, improved after the revision. However, there are still minor issues to be addressed, i.e., some "....." left along that give the impression that we received a draft; in the title, "gene variant-related" should be considered; etc.Author Response
Please see attachment

Reviewer 2 Report
Comments and Suggestions for Authors
The authors have thoroughly addressed all the reviewer's comments.
Author Response
Reviewer 2 declared that she was satisfied with the last response and no further response is therefore required.